

# Nutrient and biomass dynamics for dual-organ yield in turmeric (*Curcuma longa* L.)

Wenxin Liao[1,2,*], Haohan Wang[1,2,*], Heling Fan[1,2], Jie Chen[1], Lili Yin[1,3], Xiaoyang Cai[1,2] and Min Li[1,2]

[1] Chengdu University of Traditional Chinese Medicine, Chengdu, China
[2] Sichuan Research Center for Demonstration Project of Entire Industrial Chain of Genuine Medicinal Materials, Chengdu, China
[3] Nanchong Institute for Food and Drug Control, Nanchong, China
[*] These authors contributed equally to this work.

Corresponding author
Min Li, 028limin@163.com

## ABSTRACT

**Background**. In China, *Curcuma longa* L. is primarily cultivated for its underground parts—rhizomes (commonly known as turmeric) and tubers (Yujin), with the latter holding greater market value. However, current cultivation practices in China remain largely traditional, lacking scientific optimization in nutrient management, growth cycle alignment, or soil fertility strategies. This study aims to establish a scientific foundation for precision fertilization by investigating the dynamic patterns of dry matter accumulation and nutrient distribution in multiple plant organs throughout the growth cycle.

**Methods**. The experiment was conducted in Shuangliu, Sichuan Province, a key production area for *Curcuma longa* in China. From 55 to 209 days after planting (DAP), nine sampling points representing different phenological stages were selected. At each stage, we systematically monitored the accumulation of dry matter and the distribution of nine essential nutrient elements—nitrogen (N), phosphorus (P), potassium (K), calcium (Ca), magnesium (Mg), iron (Fe), manganese (Mn), copper (Cu), and zinc (Zn)—across five plant organs: leaves, stems, rhizomes, tubers, and roots.

**Results**. The total dry matter accumulation in *Curcuma longa* followed a typical S-shaped curve, reaching its peak at 195 DAP. Resource allocation patterns varied across four distinct growth stages. Before October (0–111 DAP), the aboveground parts dominated, with leaves and stems comprising 62.73% to 79.30% of the total dry mass. After October (111–195 DAP), underground development intensified, with priority given to the rhizomes and tubers. By late December (195 DAP), dry matter in the tubers peaked, and by early January (209 DAP), over 70% of the total dry mass was allocated below ground. Nutrient uptake also showed distinct temporal patterns. Total accumulation of nutrients in mature plants was as follows: K (1,492.39 mg), N (1,198.81 mg), P (396.98 mg), Ca (339.51 mg), Mg (210.63 mg), Fe (15.17 mg), Zn (1.15 mg), Mn (0.69 mg), and Cu (0.25 mg). The relative nutrient demand ranked as follows: K > N > P (macronutrients), Ca > Mg (secondary nutrients), and Fe > Zn > Mn > Cu (micronutrients).

**Conclusion**. The growth and development of *Curcuma longa* depend on sufficient uptake of potassium and nitrogen, moderate amounts of phosphorus, calcium, and magnesium, and trace amounts of iron, zinc, manganese, and copper—of which

potassium is required in the greatest quantity. These findings highlight the importance of adopting a stage-specific fertilization strategy to align with the plant's shifting nutrient demands throughout its life cycle.

# INTRODUCTION

*Curcuma longa* L., a perennial herb in the Zingiberaceae family, is widely cultivated across tropical and subtropical regions due to its broad economic value (*Wu & Larsen, 2000*). Global demand for turmeric has surged in recent years, driven by growing interest in natural ingredients for food and cosmetics, along with increasing recognition of its health benefits. In 2023, the global turmeric market was valued at USD 3.27 billion and is projected to reach USD 5.85 billion by 2035 (*Turmeric Market Size, 2025*). The pharmacological value of turmeric primarily stems from its curcuminoids and volatile oils. Numerous studies have confirmed that curcumin possesses strong antioxidants and anti-inflammatory properties, making it beneficial for treating various inflammatory conditions, including metabolic syndrome (*Hewlings & Kalman, 2017*). Globally, turmeric rhizomes are the primary commercial product. However, in China, both the rhizomes (*Curcumae Longae Rhizoma*, also known as turmeric) and tubers (*Curcumae Radix*, also known as Yujin) hold significant agricultural and medicinal value (*Sun et al., 2017*). According to the 2020 edition of the *Pharmacopoeia of the People's Republic of China*, Yujin is traditionally used to promote blood circulation, relieve pain and improve depression (*Ao et al., 2022*; *Chen et al., 2020b*; *Niu et al., 2024*), with a documented history of over a thousand years (*Su, 1957*). Its market value is notably high—the price of tubers is at least five times that of rhizomes—underscoring its economic importance (*Chinese Herbal Medicine World Network, 2025a*; *Chinese Herbal Medicine World Network, 2025b*). Shuangliu District in Sichuan Province is a recognized Daodi production region for medicinal turmeric and Yujin (*Chen et al., 2020a*). In traditional Chinese medicine, the concept of Daodi refers to medicinal materials cultivated in specific geographic areas with unique environmental conditions and long-established cultivation practices, believed to result in superior quality and clinical efficacy. This legacy has earned Shuangliu-grown turmeric a strong reputation and high market value, making it a key local cash crop and a vital contributor to regional agricultural income.

Despite its economic importance, cultivation in this region still largely relies on traditional farming practices, which often lack scientific nutrient management. This gap between traditional and modern agricultural approaches limits long-term profitability for local farmers. Understanding the dynamics of nutrient uptake—including macronutrients (nitrogen, phosphorus, potassium) and secondary/micronutrients—is essential for improving yield and quality (*Kaushal, 2024*; *Kumar et al., 2024*; *Paponov et al., 2023*; *Thomas et al., 2024*). These nutrients not only support biomass production but also

influence the biosynthesis of pharmacologically active compounds, thereby affecting therapeutic efficacy. Most global research on turmeric nutrition has focused on optimizing rhizome production in tropical regions such as India and Indonesia (*Amala et al., 2019*; *Nurhayati & Alhaffiz, 2024*). There is limited attention to tuber development or adaptation to subtropical climates like Shuangliu, resulting in a knowledge gap regarding the specific cultivation needs of dual-purpose turmeric production. Moreover, differences in soil composition, cultivars, and agro-climatic conditions across regions may lead to varied nutrient requirements (*Wang et al., 2022*), highlighting the need for localized, systematic studies to guide evidence-based fertilization strategies.

This study aims to investigate the cultivation of *Curcuma longa* in Shuangliu, focusing on nutrient uptake, distribution, and accumulation in different plant organs throughout its growth stages. The goal is to provide a scientific basis for stage-specific fertilization strategies. These strategies are essential for improving the yield and quality of both rhizomes and tubers, enhancing resource use efficiency, and supporting the sustainable industrial development of turmeric and Yujin in the region.

## MATERIALS AND METHODS

### Basic situation of yield

The experimental site is located at the turmeric planting base in Shuangliu District, Chengdu, Sichuan Province, China. This region has a subtropical humid monsoon climate with an altitude of 495 m above sea level. The average annual temperature is about 15 °C, and the average annual sunshine duration is over 1,000 h. The average annual precipitation is 945.6 mm, with most of the rainfall occurring in July and August. The frost-free period lasts approximately 275 days, with the first frost typically occurring in late November (167 days) and the last frost in late February. The test soil type is sandy loam with a pH of 7.05. The soil contains 45.4 mg/kg of alkaline nitrogen (N), 65.94 mg/kg of available phosphorus (P), 94.56 mg/kg of available potassium (K), and 26.80 g/kg of organic matter.

### Experimental design

The experimental turmeric seed rhizomes were obtained from the officially released new cultivar 'Chuanjianghuang NO.1'. Seed rhizomes (6 kg each) were prepared and sown on June 16, 2019, 0 days after planting (DAP). Each plot had an area of 30 m$^2$ (10 m × 3 m), with one m wide walkways between plots and a plant spacing of 35 cm × 35 cm. In late July, fertilizer was applied after seedlings emerged. A standard compound fertilizer (N-P$_2$O$_5$-K$_2$O 15: 15 :15) was applied, and the amount of fertilizer was 100 kg per 667 m$^2$. The experimental design included three replications, with cultivation management practices (*Qu et al., 2022*).

Throughout the growing season, ten marked plants per plot were monitored using non-destructive measurements. Destructive sampling was carried out at nine distinct time points: August 9 (55 days after planting, DAP), August 23 (69 DAP), September 6 (83 DAP), October 4 (111 DAP), October 18 (125 DAP), November 1 (139 DAP), November 29 (167 DAP), December 27 (195 DAP), and January 10 of the following year (209 DAP). At each time point, ten plants were randomly selected and sampled between 9:00 and 10:00

a.m. In addition, soil samples (0.5 kg) were collected from the root zone of the sampled plants, labeled on-site, and immediately placed in sample bags for transport to the Chinese Medicinal Resources and Identification Laboratory, Chengdu University of Traditional Chinese Medicine, for further processing.

## Determination index and method

In each plot, 10 plants were marked according to the five-point sampling method for observation, and growth indicators such as plant height, leaf length, leaf width, stem diameter, and leaf number were recorded during different growth stages. Plant height (distance from the top leaf to the root-shoot base), leaf length, and leaf width were measured using a tape measure. Stem diameter was measured approximately two cm from the root base using a ruler. The number of leaves was counted, excluding those with more than 50% wilting. The sampled plants were washed with distilled water and dried by blotting. They were then separated into five parts: leaf, stem, rhizome, adventitious roots, and tuber root, as shown in Fig. 1. The fresh weight of each part was recorded. The plant samples were then blanched at 105 °C for 30 min in a forced-air drying oven and dried at 55 °C to constant weight. The dry weight of each organ was recorded to calculate the dry matter accumulation, daily increase, and distribution rate for each growth stage. After drying, the samples were ground and sieved through a 0.5 mm mesh. Total nitrogen was measured using the Kjeldahl method (*Sparks et al., 1996*), total phosphorus using the molybdenum-antimony colorimetric method, total potassium by flame photometry (*Kalra & Soil and Plant Analysis Council, 1998*), and calcium, magnesium, iron, manganese, zinc, and copper using ICP-MS (*Mittal et al., 2017*).

## Data analysis

Data were organized and analyzed using Excel 2019 (Microsoft, Redmond, WA, USA) and SPSS 21.0 (IBM Corp., Armonk, NY, USA). One-way analysis of variance (ANOVA) was used to compare differences among groups. When homogeneity of variance was assumed, Duncan's multiple range test was applied for pairwise comparisons (*Azis et al., 2024*); when variances were unequal, Tamhane's T2 test was used. A $p$-value $\leq 0.05$ was considered statistically significant.

# RESULTS

## Dynamic accumulation of dry matter in turmeric plants

The total dry matter accumulation in *Curcuma longa* followed a characteristic S-shaped growth curve, peaking at 195 days after planting (DAP), with a slight decline thereafter (Fig. 2A). During the early growth stage (55–111 DAP), the aboveground parts—leaves and stems—dominated biomass allocation, accounting for 62.73% to 79.30% of the total dry weight (Fig. 2B). Leaf biomass reached its maximum around 125 DAP. By this stage, dry matter distribution between aboveground and belowground parts had become relatively balanced, with the highest proportion shifting to the rhizomes (34.05%). Stem biomass peaked at 139 DAP. Following this period, dry matter allocation shifted significantly toward the below ground organs. Both rhizomes and tubers showed rapid biomass accumulation,

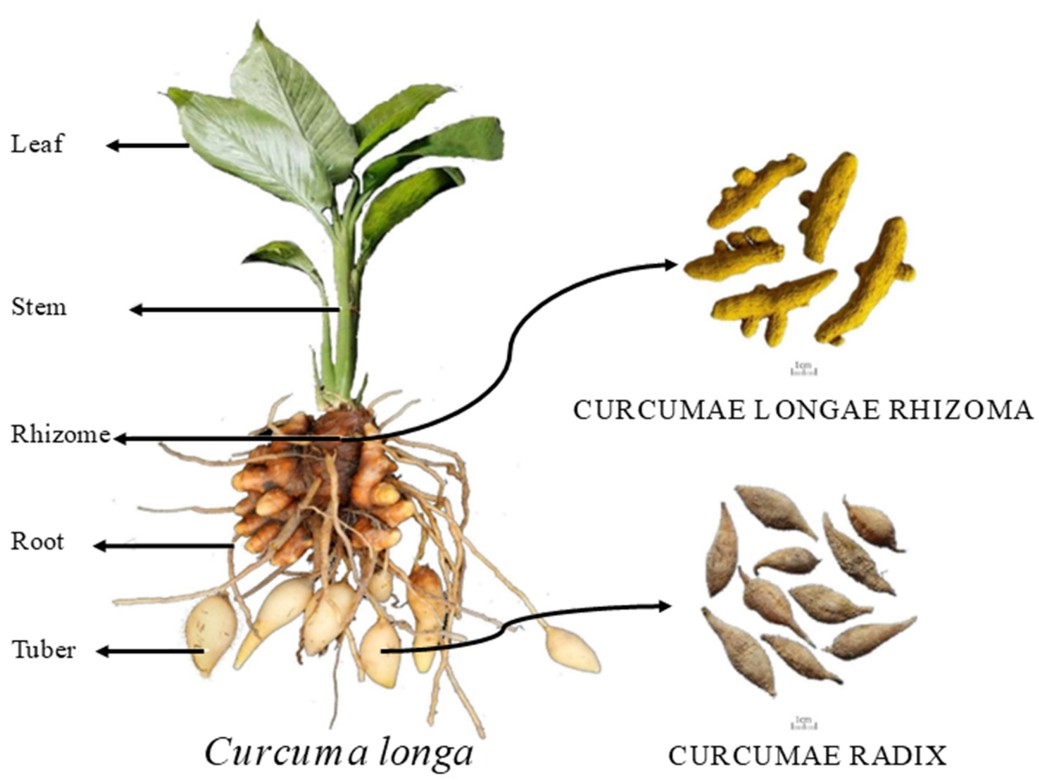

**Figure 1** Schematic diagram of turmeric plant.

with tubers reaching their peak at 195 DAP. By the end of the growing season (195–209 DAP), more than 70% of the total dry matter was concentrated in the belowground organs—rhizomes, tubers, and roots—indicating a substantial translocation of assimilates into storage structures.

## Overall nutrient accumulation and demand hierarchy in turmeric plants

Across its entire growth cycle, turmeric accumulated total nutrients in the following order: K (1,492.39 mg), N (1,198.81 mg), P (396.98 mg), calcium (Ca) (339.51 mg), magnesium (Mg) (210.63 mg), iron (Fe) (15.17 mg), zinc (Zn) (1.15 mg), manganese (Mn) (0.69 mg), and copper (Cu) (0.25 mg). This established a nutrient demand hierarchy of K > N > P for macronutrients, Ca > Mg for secondary nutrients, and Fe > Zn > Mn > Cu for micronutrients.

## Macronutrient dynamics (N, P, K)
### Nitrogen

Leaf nitrogen content showed an initial slow increase from 55 to 83 DAP, followed by a slight decline, then a rapid increase to a maximum at 125 DAP, and a gradual decrease thereafter. Stem nitrogen continuously increased from 55 to 125 DAP, peaking at 2.969%, then rapidly decreased and stabilized. Rhizome nitrogen slowly decreased from 55 to 125 DAP before gradually rising. Tuber nitrogen was highest at its initial formation (111 DAP)

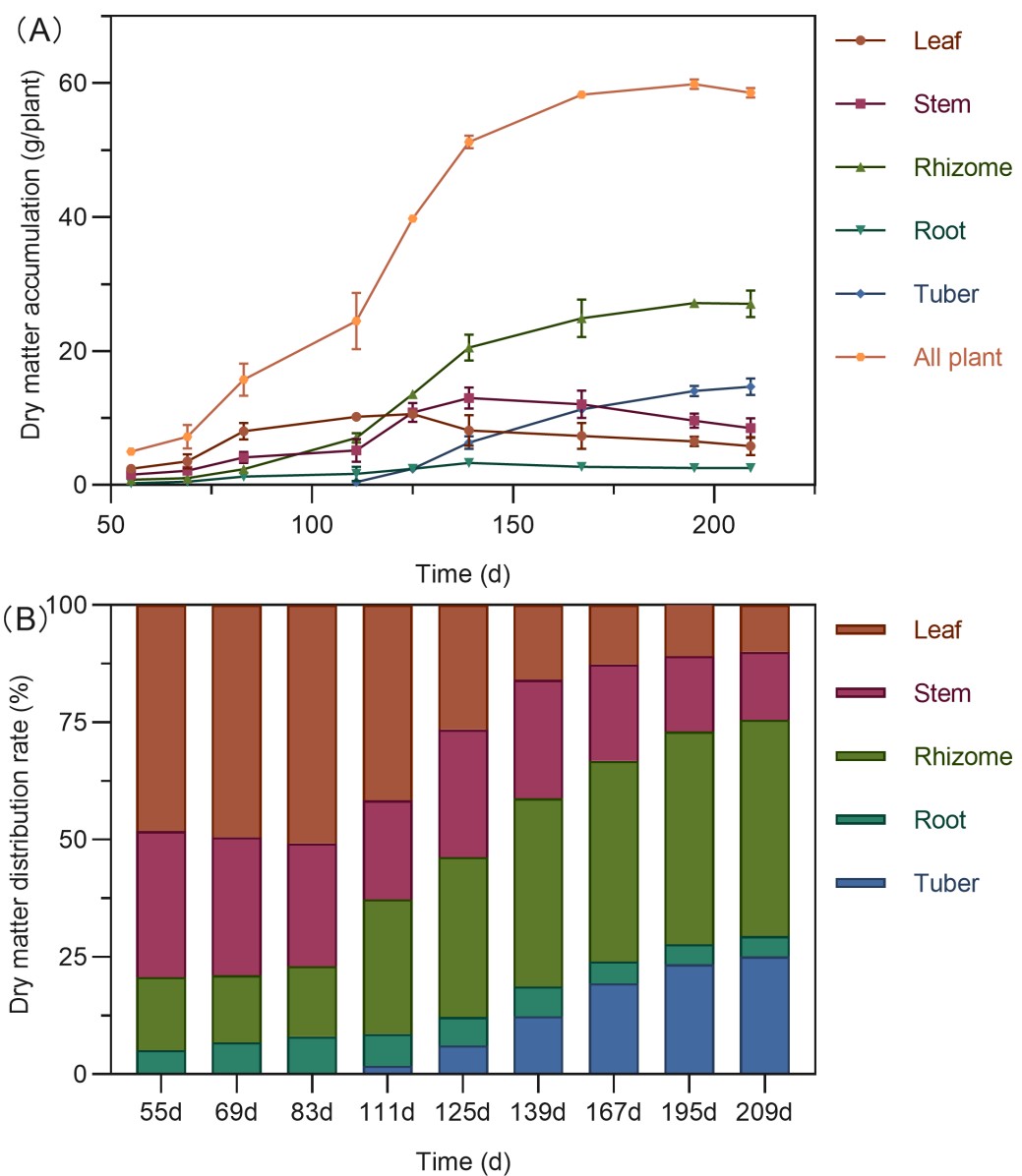

**Figure 2  Dry matter accumulation (A) and distribution (B) in turmeric plants.**

and then progressively decreased and stabilized (Fig. 3A). In terms of accumulation, above-ground leaf and stem nitrogen continuously increased from 55 to 125 DAP, reaching maximums of 303.587 mg and 320.054 mg, respectively, before declining as the plant matured. For underground parts, rhizome and tuber nitrogen accumulation increased and then stabilized, with the rhizome exhibiting the highest nitrogen accumulation capacity, exceeding 599 mg by 195 DAPS. Root nitrogen remained low throughout. At the end of the growing season, nitrogen accumulation ranked: rhizome > stem > tuber > leaf > root (Fig. 3B). Regarding distribution, nitrogen allocation to above-ground leaves and stems progressively decreased, while allocation to underground rhizomes and tubers steadily

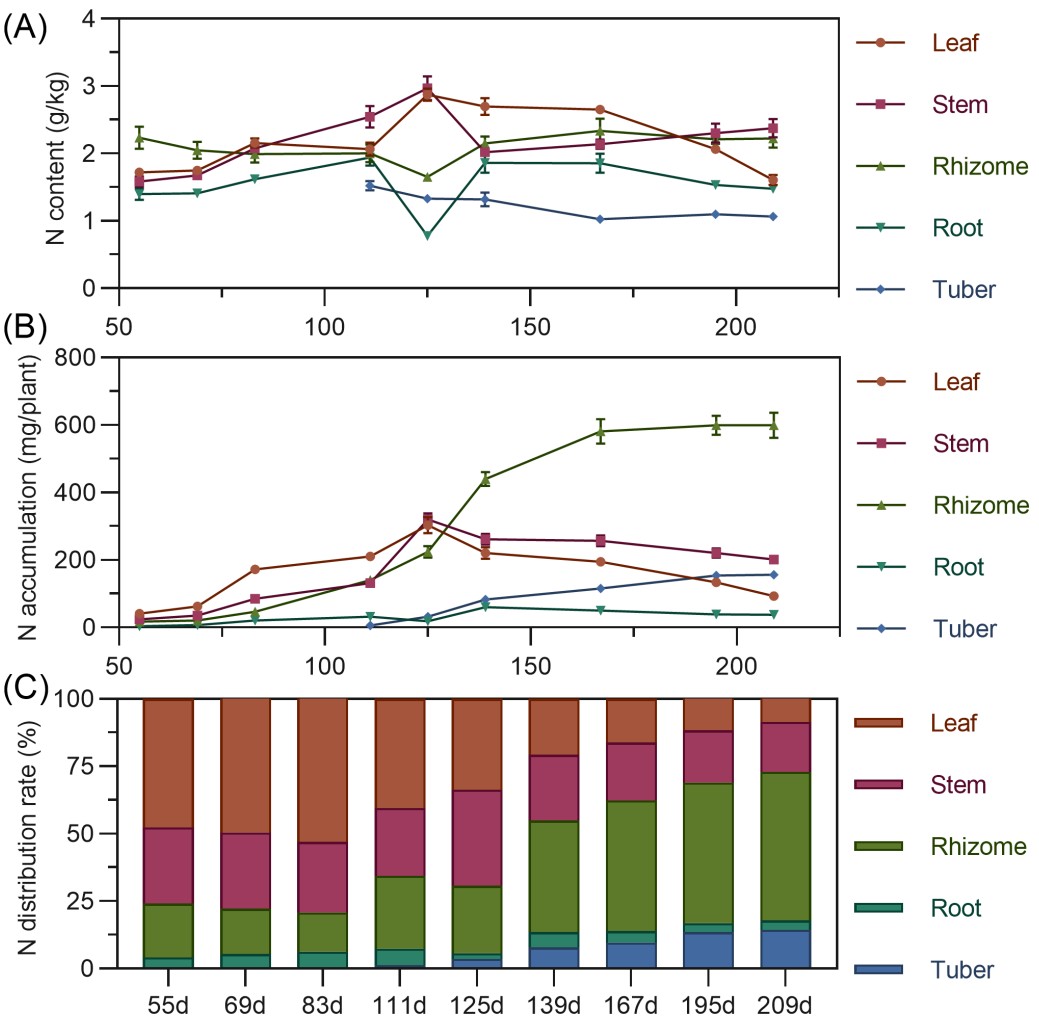

**Figure 3** (A) Changes in nitrogen content in turmeric. (B) Dynamic changes in nitrogen accumulation in turmeric plants. (C) Changes in the nitrogen allocation rate in turmeric plants.

increased. From 55 to 125 DAP, nitrogen was predominantly allocated to above-ground parts (65.63% to 79.22%). However, from 139 DAP, the nitrogen distribution significantly shifted to underground organs, reaching a total distribution rate of 72.90% by 209 DAP, with the rhizome consistently having the highest allocation (>40%) and the tuber being secondary (Fig. 3C).

### Phosphorus

Leaf phosphorus content exhibited a complex trend, initially decreasing, then increasing to a second peak at 139 DAP, before gradually declining. Stem phosphorus fluctuated minimally from 55 to 139 DAP, then rapidly increased between 139 and 167 DAP to reach its maximum, followed by a gradual decrease. Rhizome phosphorus content initially decreased, then increased, fluctuating before stabilizing after 167 DAP. Tuber phosphorus content was highest at its initial formation (111 DAP), progressively decreasing and

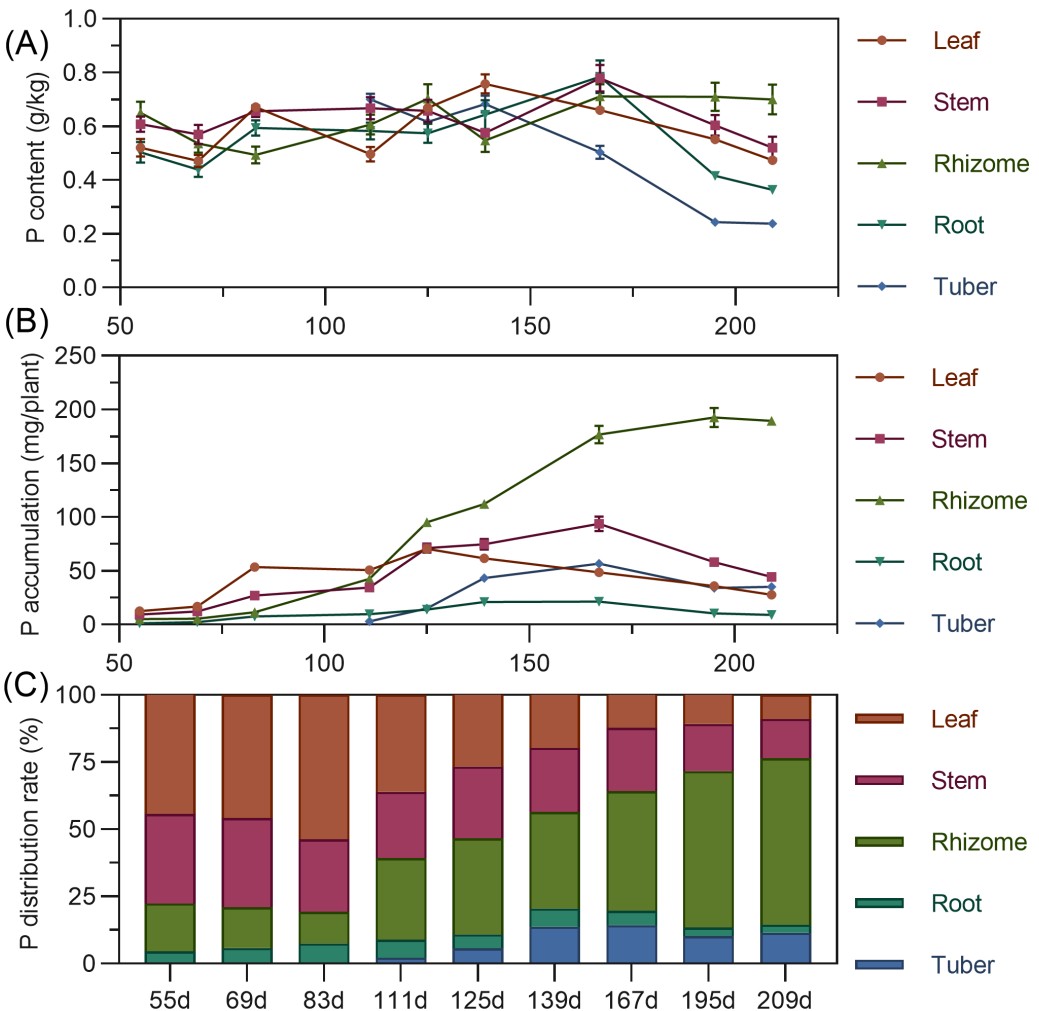

**Figure 4** (A) Changes in phosphorus content in turmeric plants. (B) Dynamic changes in phosphorus accumulation in turmeric plants. (C) Changes in the phosphorus allocation rate in turmeric plants.

stabilizing after 195 DAP (Fig. 4A). For accumulation, above-ground leaf phosphorus slowly rose from 55 to 125 DAP, peaking before a gradual decline. Stem phosphorus continuously increased from 55 to 167 DAP, reaching maximum accumulation, then gradually declined. In the underground parts, rhizome phosphorus steadily increased, peaking at 192.689 mg by 195 DAP. Root and tuber phosphorus rose then stabilized, with maximum values (root 21.43 mg, tuber 56.754 mg) at 167 DAP (Fig. 4B). In terms of distribution, above-ground leaf and stem phosphorus allocation gradually decreased, while underground rhizome allocation continually increased. Before 111 DAP, phosphorus was predominantly above-ground. By 125 DAP, above-ground and underground distribution were approximately 50% each. After this, the distribution clearly shifted to underground parts, reaching 76.47% by 209 DAP, with the rhizome having the highest distribution (Fig. 4C).
### Potassium

Leaf potassium content gradually decreased from 55 to 125 DAP, then increased to a peak at 139 DAP, before declining and stabilizing. Stem showed high potassium content early (55 DAP, 6.432%), then a steady decline from 55 to 125 DAP, followed by stabilization. Rhizome potassium content fluctuated downward throughout. Tuber had the highest potassium content at 111 DAP, then progressively decreased (Fig. 5A). Regarding accumulation, above-ground leaf potassium peaked at 111 DAP (290.400 mg), and stem at 139 DAP (420.477 mg), then both decreased. In the underground parts, rhizome and root potassium increased then gradually decreased, with the rhizome peaking at 167 DAP. Tuber steadily accumulated potassium from 111 DAP. At the end of the growth period, potassium accumulation ranked: rhizome > tuber > stem > leaf > root (Fig. 5B). For distribution, above-ground leaf and stem potassium allocation decreased, while underground rhizome and tuber allocation steadily increased. From 55 to 125 DAP, above-ground total potassium distribution was higher. By 139 DAP, the distribution center began to shift to underground parts, mainly the rhizome (Fig. 5C).

## Secondary nutrient dynamics (Ca, Mg)
### Calcium

Calcium content in above-ground leaves and stems continuously increased throughout the growth period, stabilizing after 195 DAP. In contrast, underground rhizome, root, and tuber calcium content fluctuated downward, remaining relatively low throughout. In later stages, content ranked: leaf > stem > root > rhizome > tuber (Fig. 6A). In terms of accumulation, above-ground leaf and stem calcium continuously increased throughout. Underground rhizome, root, and tuber calcium fluctuated within a smaller range. By later stages, above-ground calcium accumulation far exceeded underground (Fig. 6B). Regarding distribution, nearly all calcium was allocated to above-ground parts (86.40% to 97.30% in leaf and stem). Underground rhizome, root, and tuber calcium distribution was below 10%, with the tuber lowest (Fig. 6C).

### Magnesium

Leaf magnesium content gradually decreased from 55 to 111 DAP, then rapidly increased, reaching its highest value by 167 DAP. Stem magnesium rapidly increased from 55 to 111 DAP, peaking at 8.698 g/kg, then gradually decreased. Underground rhizome and root had the highest magnesium content early, then steadily decreased over time. Tuber had the highest magnesium content at initial formation (111 DAP), then gradually decreased (Fig. 7A). For accumulation, leaf magnesium showed a fluctuating upward trend, peaking at 167 DAP before gradual decline. Stem accumulated magnesium continuously from 55 to 139 DAP, peaking at 95.539 mg, then rapidly declined. Underground rhizome, root, and tuber gradually increased in magnesium accumulation, reaching maximum values before decreasing. By the end of the growth period, rhizome had the highest magnesium accumulation, followed by stem, with root lowest (Fig. 7B). Regarding distribution, from 55 to 167 DAP, above-ground magnesium distribution was higher. Starting at 195 DAP,

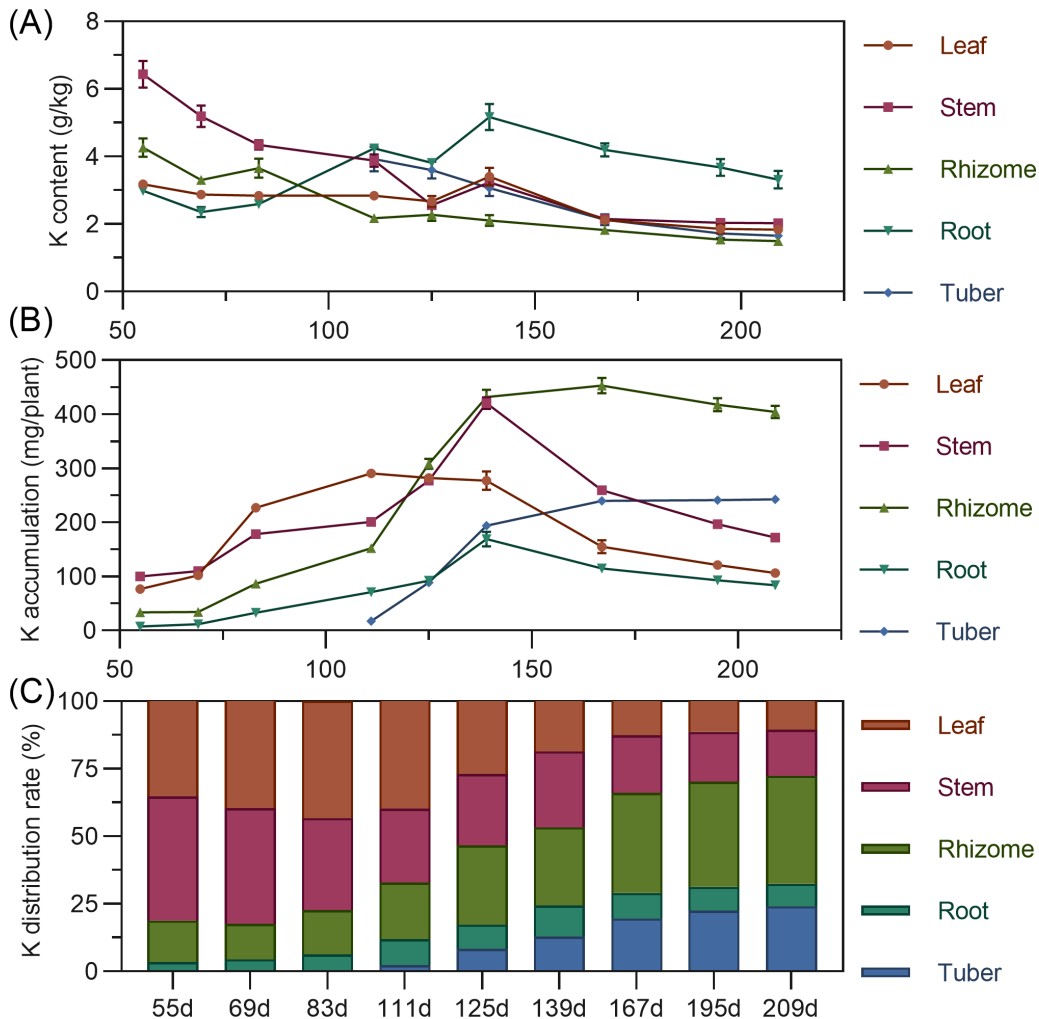

**Figure 5** (A) Changes in potassium content in turmeric plants. (B) Dynamic changes in potassium accumulation in turmeric plants. (C) Changes in the potassium allocation rate in turmeric plants.

the distribution center shifted to underground parts, with the rhizome receiving the highest allocation (>40%) (Fig. 7C).

## Micronutrient dynamics (Fe, Mn, Cu, Zn)
### Iron

Iron content in above-ground leaves and stems remained relatively stable early, then rapidly increased later. Leaf iron was stable from 55 to 139 DAP, then increased linearly. Stem iron was stable from 55 to 167 DAP, then increased sharply. Underground, rhizome iron was highest at 55 DAP, then continuously decreased, approaching zero. Tuber iron gradually increased from 111 to 125 DAP, peaked at 139 DAP, then rapidly declined. In later stages, leaf iron was highest, followed by stem, with underground organs relatively low (Fig. 8A). In terms of accumulation, organ iron was low early; later, above-ground leaf and stem far surpassed underground. Leaf iron fluctuated and slowly increased from

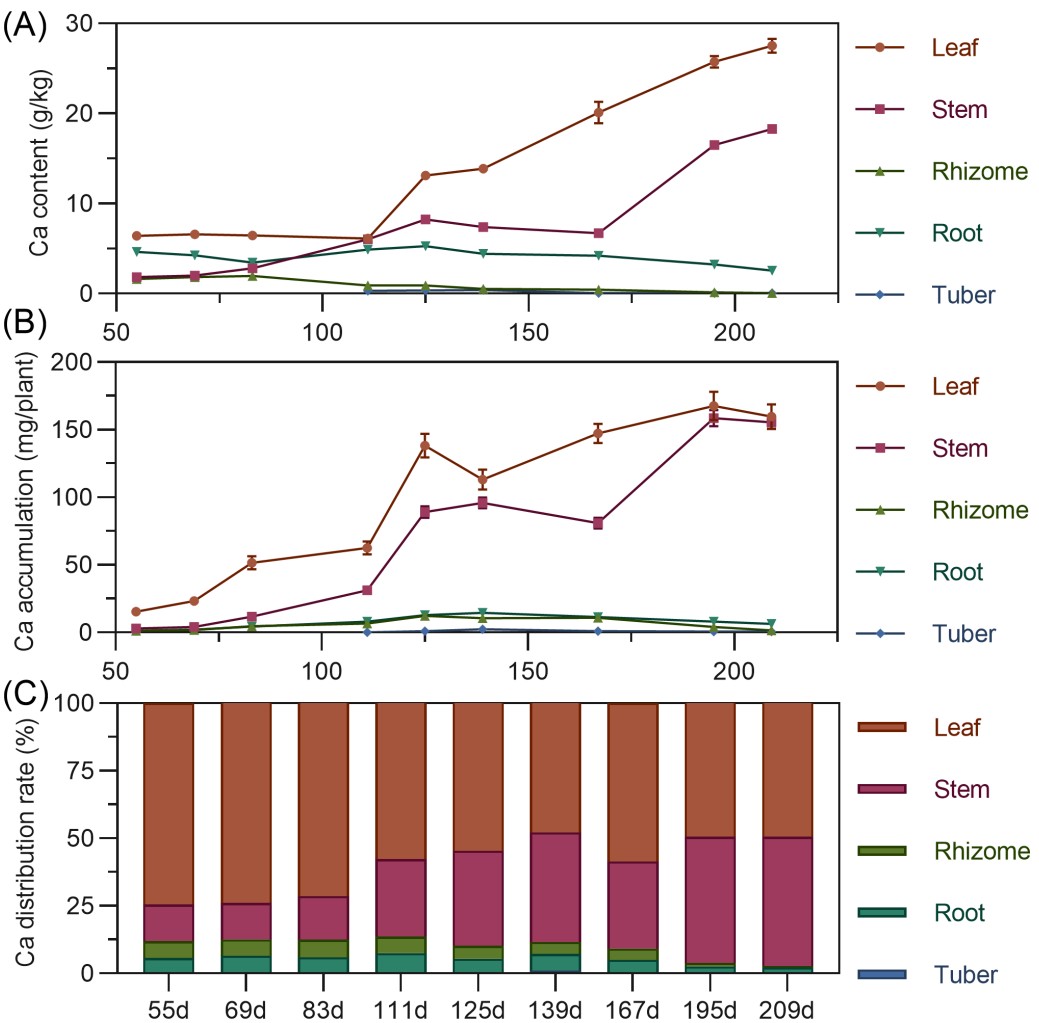

**Figure 6** (A) Changes in calcium content in turmeric plants. (B) Dynamic changes in calcium accumulation in turmeric plants. (C) Changes in the calcium allocation rate in turmeric plants.

55 to 139 DAP, then rose sharply. Stem iron slightly increased from 55 to 167 DAP, then decreased, then linearly increased after 167 DAP, peaking at 7.66 mg at 195 DAP (Fig. 8B). Regarding distribution, iron's distribution center shifted with stages. From 55 to 69 DAP, it was mainly in the leaf (44.66% to 46.21%). It then shifted to the root, and by 167 DAP, shifted back to the leaf. From 195 to 209 DAP, the distribution was mainly in above-ground leaf and stem, with total exceeding 95% (Fig. 8C).

### Manganese

Manganese content in above-ground leaves and stems showed a fluctuating upward trend (leaf >stem). Conversely, underground rhizome, root, and tuber manganese was highest early, then gradually decreased. In later stages, leaf manganese was highest, followed by stem, while rhizome and tuber manganese was extremely low (Fig. 9A). For accumulation, above-ground leaf and stem manganese increased fluctuatingly. Underground rhizome

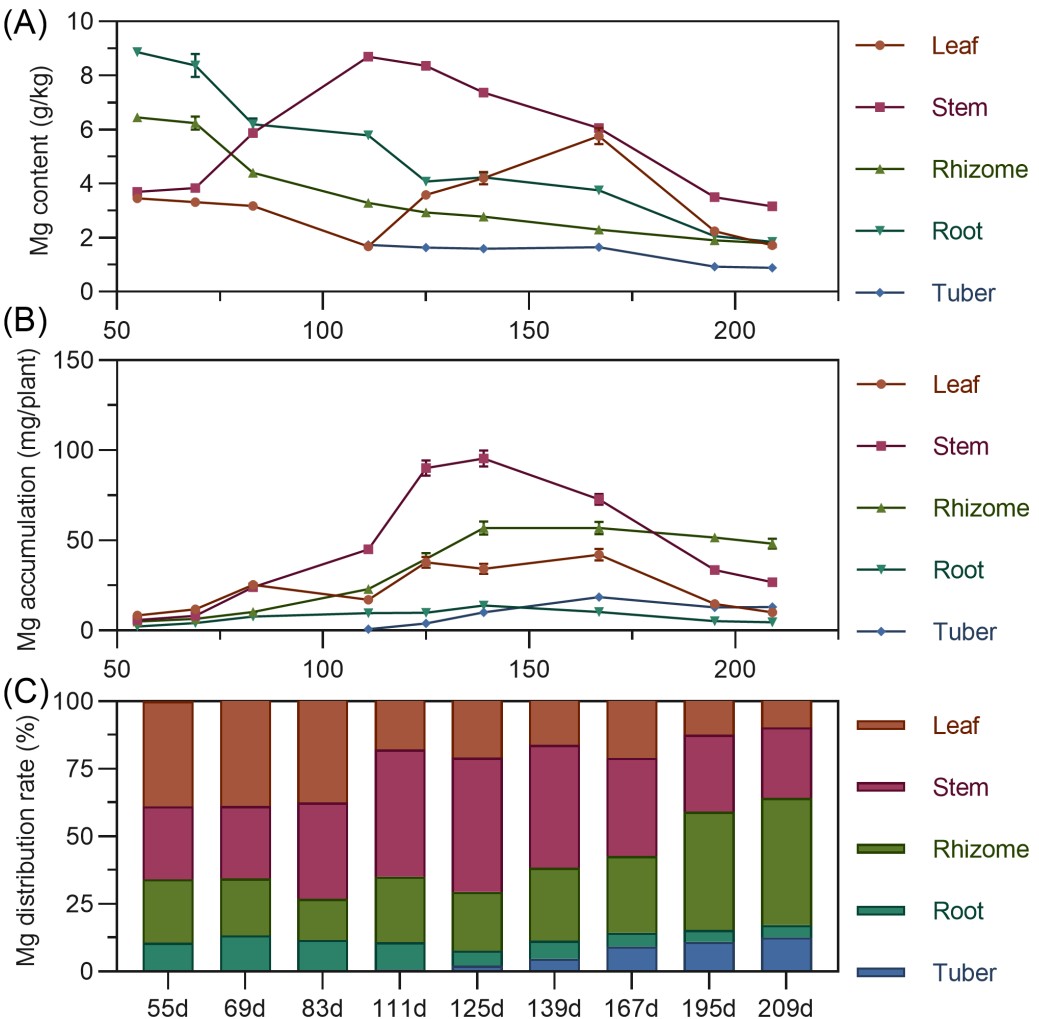

**Figure 7** (A) Changes in magnesium content in turmeric plants. (B) Dynamic changes in magnesium accumulation in turmeric plants. (C) Changes in the magnesium allocation rate in turmeric plants.

and root accumulation initially increased, then decreased. Leaf consistently had the highest manganese, while tuber exhibited extremely low accumulation (Fig. 9B). Manganese distribution consistently remained in above-ground parts (56.38% to 89.11%) throughout. Leaf had the highest manganese distribution rate (>40%) (Fig. 9C).

### Copper

Copper content in above-ground leaves and stems gradually increased with minor fluctuations. Rhizome copper increased from 55 to 83 DAP, then decreased and stabilized. Tuber copper initially increased, then decreased, remaining low. Leaf copper accumulation showed a bimodal curve, rising from 55 to 111 DAP to a peak, then decreasing; after 139 DAP, it increased again to a second peak before decreasing. Stem copper continuously increased throughout, with a slight decline after 195 DAP (Fig. 10A). Underground rhizome and tuber gradually increased in copper accumulation, peaking at 167 DAP, then slightly

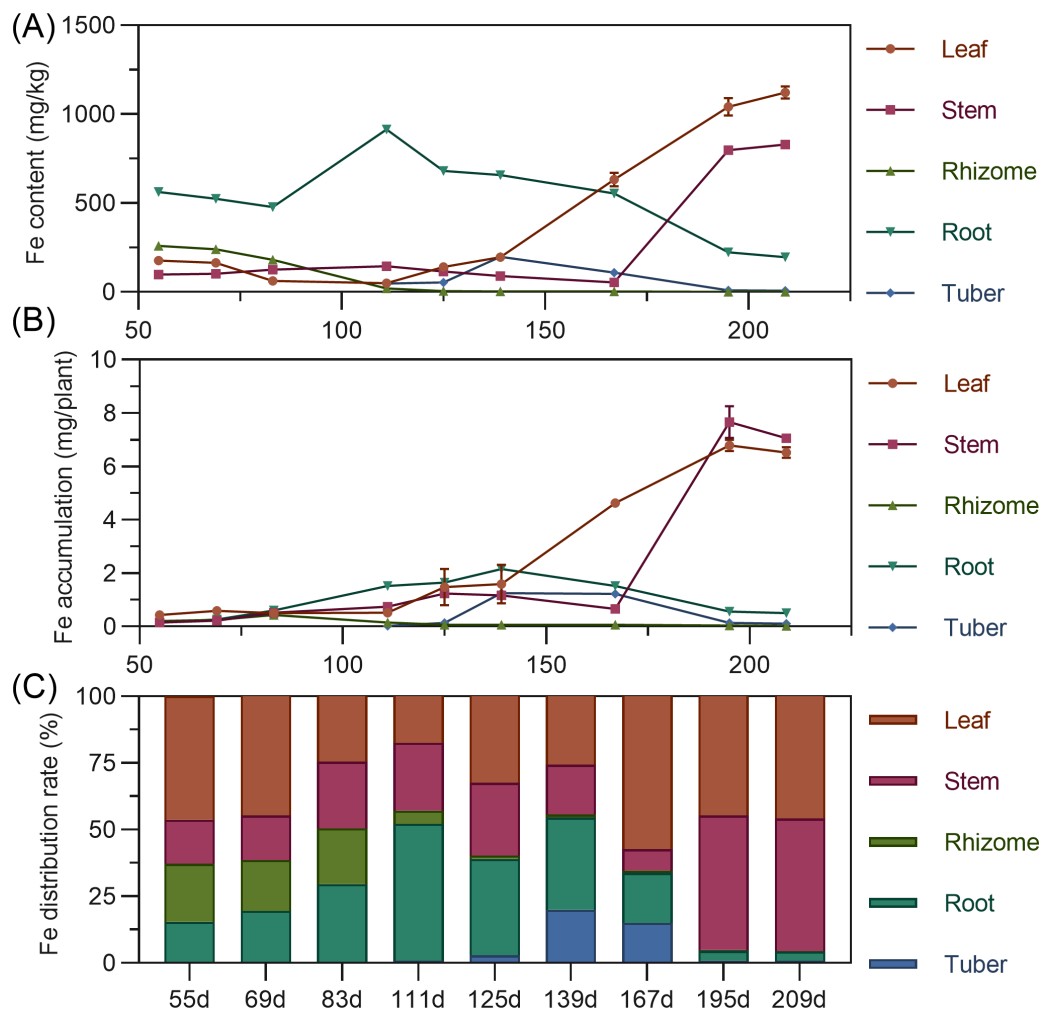

**Figure 8** (A) Changes in iron content in turmeric plants. (B) Dynamic changes in iron accumulation in turmeric plants. (C) Changes in the iron allocation rate in turmeric plants.

decreased. In later stages, copper accumulation ranked: rhizome > stem > root > leaf > tuber (Fig. 10B). Copper distribution was in above-ground parts until around 69 DAP, then shifted to underground parts (Fig. 10C).

*Zinc*

Leaf zinc content gradually increased throughout. Stem zinc decreased from 55 to 125 DAP, peaked at 125 DAP, then declined, and rapidly increased after 167 DAP. Underground rhizome and tuber had the highest zinc early in formation, then gradually decreased (Fig. 11A). For accumulation, leaf zinc gradually increased and stabilized after 195 DAP. Stem zinc fluctuated, rising from 55 to 125 DAP to a peak, then decreasing, and increasing again after 167 DAP. Rhizome zinc fluctuated most significantly, with a rapid increase and a peak at 195 DAP. In later stages, stem > rhizome > leaf > root > tuber. Zinc was primarily concentrated in the stem and rhizome (Fig. 11B). In above-ground parts, stem

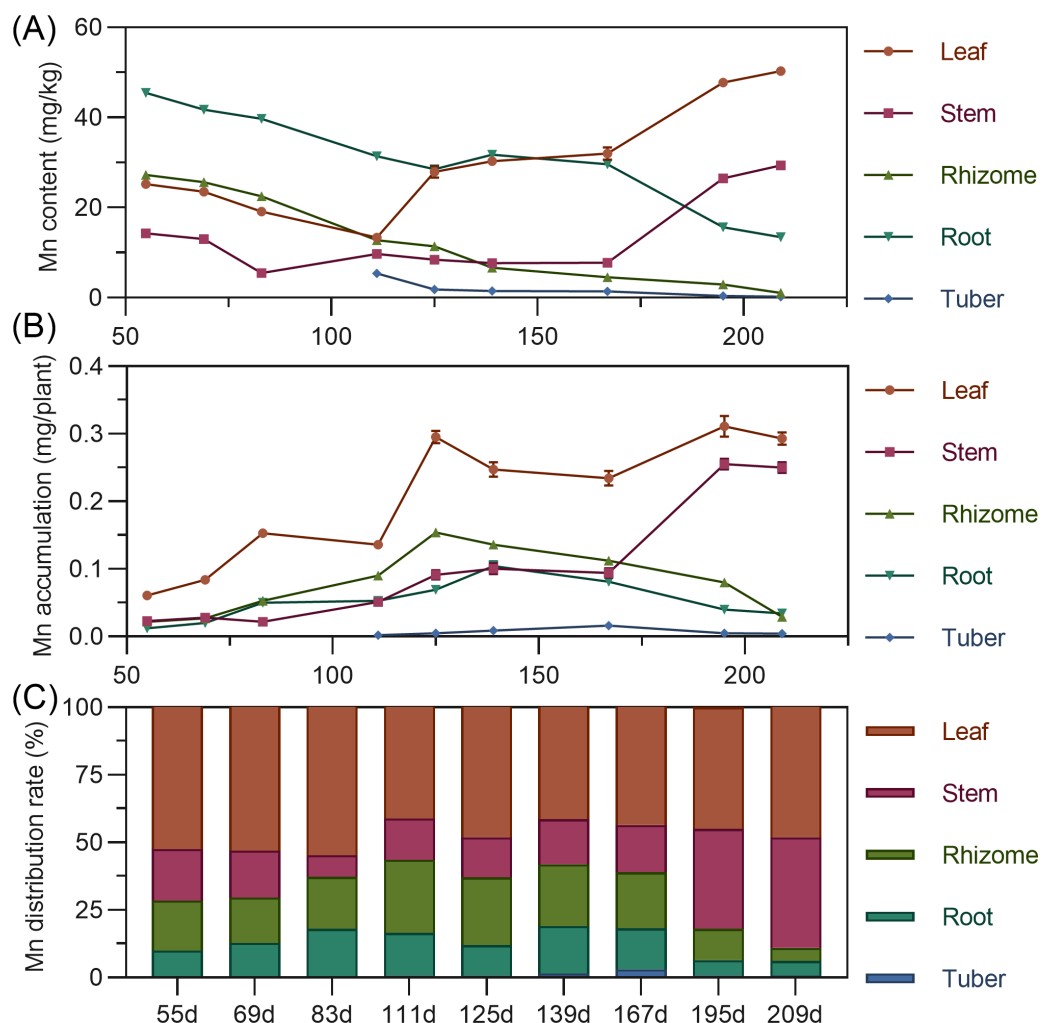

**Figure 9** (A) Changes in manganese content in turmeric plants. (B) Dynamic changes in manganese accumulation in turmeric plants. (C) Changes in the manganese allocation rate in turmeric plants.

zinc distribution was higher than leaf. In underground parts, rhizome had the highest zinc distribution, followed by root, with tuber lowest (Fig. 11C).

## DISCUSSION

### A phenological framework for understanding nutrient demand in turmeric

This study revealed a classic S-shaped growth curve in the total dry matter accumulation of *Curcuma longa*, with biomass peaking at 195 days after planting (DAP) followed by a slight decline. More importantly, the results identified a critical physiological turning point between 111 and 125 DAP. Prior to this transition, plant growth was primarily focused on the development of aboveground vegetative organs—leaves and stems—which accounted for 62.73% to 79.30% of total dry matter. After this point, the plant actively redirected

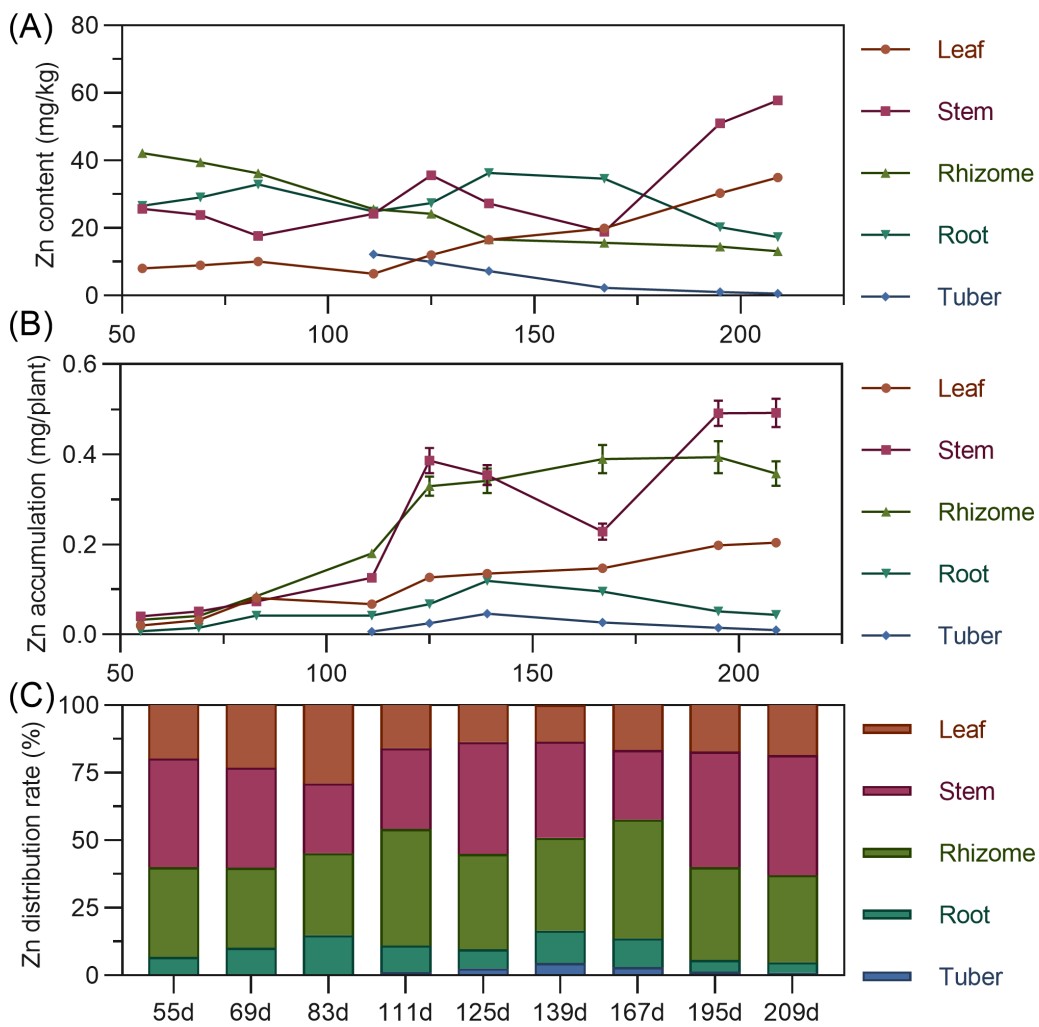

**Figure 10 (A) Changes in zinc content in turmeric plants. (B) Dynamic changes in zinc accumulation in turmeric plants. (C) Changes in the zinc allocation rate in turmeric plants.**

resources toward belowground storage organs. By the end of the growth cycle (209 DAP), rhizomes, tubers, and roots together comprised more than 70% of the total plant biomass.

By translating growth dynamics into measurable data, we propose a refined five-stage phenological model tailored to the subtropical growing conditions of Shuangliu, China. This model offers a precise framework for managing *Curcuma longa* cultivation: (1) Seedling Stage (0–69 DAP): focused on germination and early development of roots and shoots; (2) Leaf rosette and initial rhizome expansion (69–111 DAP): rapid aboveground growth enhances photosynthetic capacity, accompanied by the onset of primary rhizome formation; (3) Tuber expansion and secondary rhizome enlargement (111–139 DAP): A critical transition phase marked by a shift in resource allocation toward belowground sinks, with the initiation of tuber development and accelerated rhizome growth; (4) Dry matter accumulation in tuber and rhizome (139–195 DAP): peak swelling phase characterized by

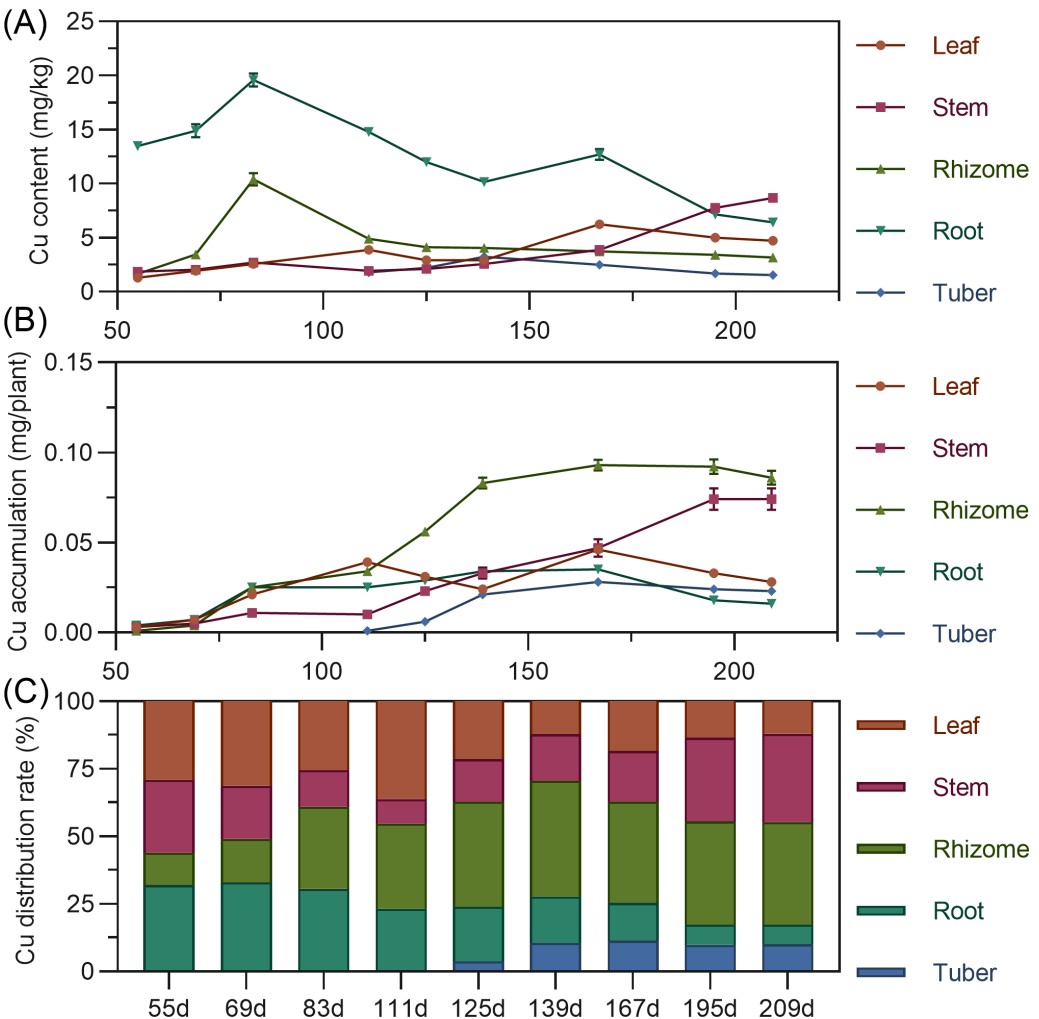

**Figure 11** (A) Changes in copper content in turmeric plants. (B) Dynamic changes in copper accumulation in turmeric plants. (C) Changes in the copper allocation rate in turmeric plants.

rapid biomass accumulation in economically important storage organs; (5) Maturation stage (195–209 DAP): physiological maturity is reached, net nutrient uptake from soil ceases, and the plant prepares for dormancy.

Compared with earlier classification models—such as the five-stage system developed for Ethiopian turmeric, which primarily focused on vegetative growth and senescence—this model provides a more detailed and functionally relevant framework (*Mekonnen & Garedew, 2019*). It clearly distinguishes between the initiation and expansion phases of both rhizomes and tubers, representing a significant improvement. This distinction is particularly important in cultivation systems like China's, where both organs possess considerable economic value.

## The roles of macronutrients in turmeric growth

The results of this study indicate that the macronutrient demand hierarchy for *Curcuma longa* is potassium (K) > nitrogen (N) > phosphorus (P), consistent with a foundational nutrient dynamics study on turmeric conducted in 1996 (*Zhang, Li & Liao, 1996*). Potassium (K): potassium is the most required nutrient throughout the entire growth cycle, with a total accumulation of 1,492.39 mg per plant. Accumulation of potassium in rhizomes and tubers sharply accelerates after 125 days after planting (DAP), coinciding with their rapid expansion phase. By harvest, these underground organs represent the major potassium sinks, underscoring potassium's critical role in their development. Such a high potassium demand is characteristic of high-yielding root and tuber crops. In plants, potassium plays key functional roles, including the transport of photosynthates from leaves to storage organs *via* the phloem, activation of over 60 enzymes, regulation of stomatal opening, and formation of storage organ yield (*Wang et al., 2013*). Research on potatoes has shown that potassium is essential for transporting sugars to developing tubers, with deficiency causing sugar accumulation in leaves and reduced tuber yield (*Torabian et al., 2021*). Members of the Zingiberaceae family, including turmeric and ginger, are known for high nutrient consumption, particularly potassium (*Jabborova et al., 2021*). The substantial potassium removal in harvested organs also means that each crop cycle extracts large amounts of potassium from the soil, classifying turmeric as a "nutrient mining" crop, especially with respect to potassium. In continuous cultivation systems, failure to replenish this potassium will inevitably deplete soil nutrient reserves and lead to declining productivity over time (*Kandasamy et al., 2012*). Therefore, this study provides not only guidance for single-season fertilization but also important insights for long-term soil fertility management and sustainable turmeric cultivation.

Nitrogen (N) and phosphorus (P): nitrogen and phosphorus are fundamental to plant growth processes. Our results show that nitrogen accumulation peaks in leaves and stems during early growth stages (around 125 DAP), reflecting its central role in synthesizing proteins and chlorophyll to support aboveground biomass development. As the plant transitions to reproductive and storage phases, accumulated nitrogen is gradually remobilized to developing rhizomes, which become the primary nitrogen sink by the end of the growing season. Phosphorus accumulation follows a similar pattern, with a marked shift from aboveground to belowground organs after 125 DAP. Notably, the peak demand for nitrogen precedes that of potassium. Stage-specific fertilization: early-stage fertilization should focus on supplying nitrogen to support aboveground growth, followed by increased potassium application during later stages to facilitate sugar transport into storage organs. This phased fertilization strategy can enhance nutrient use efficiency and optimize yield.

## Partitioning of secondary and micronutrients

Calcium (Ca), manganese (Mn), and iron (Fe): calcium, manganese, and iron primarily accumulate in the leaves and stems, with minimal translocation to belowground organs. This limited mobility is due to their low phloem mobility within the plant's transport system. Consequently, these nutrients require continuous supply from the soil throughout the growing season to maintain the health of the aboveground tissues. Declining aboveground

health during the later stages (139–195 DAP) can reduce photosynthetic capacity and ultimately limit the yield of rhizomes and tubers. Magnesium (Mg), copper (Cu), and zinc (Zn): in contrast to the relatively immobile nutrients, magnesium, copper, and zinc exhibit significant phloem mobility. Our study shows notable redistribution of these nutrients between aboveground and belowground organs during the later growth stages. Particularly, rhizomes become the main accumulation site for magnesium by the end of the growth cycle, and also serve as important sinks for copper and zinc.

Importantly, the nutritional status of medicinal plants directly influences the biosynthesis of secondary metabolites (*El-Hawaz et al., 2018*). Plants in the Zingiberaceae family are renowned for their rich active compounds, such as curcuminoids in turmeric and gingerols and shogaols in ginger, which determine their medicinal properties and market value (*Mao et al., 2019*). Magnesium, copper, and zinc are precisely transported to rhizomes where they function as enzyme cofactors in complex biosynthetic pathways. These pathways convert primary metabolites (such as sugars) into stored starch and high-value secondary metabolites like curcuminoids. For crops like *Curcuma longa*, whose value depends not only on yield but also on the content of bioactive compounds, optimizing micronutrient fertilization is a critical factor for maximizing economic benefits.

## Critical factors and nutrient requirements for tuber formation

A unique contribution of this study is the separate quantification of nutrient dynamics in tubers (referred to as Yujin in traditional Chinese medicine), which hold a significantly higher market value than rhizomes in China—priced at least five times greater. Our data indicate that the critical developmental stages of tubers—namely the expansion phase (111–139 DAP) and dry matter accumulation phase (139–195 DAP)—exhibit distinct and elevated demands for potassium, phosphorus, magnesium, copper, and iron.

Additionally, our observations reveal that tuber formation is strongly influenced by soil texture and moisture conditions. These roots preferentially develop in sandy loam soils and are prone to cracking under waterlogged conditions. This finding, combined with existing literature, suggests that tuber development may serve as an adaptive survival response to drought stress (*Yin et al., 2022*), while being inhibited by waterlogging (*Aslam et al., 2023*). These insights provide a foundation for targeted cultivation practices, such as optimized irrigation management and site-specific fertilization, aimed at improving tuber yield.

## Practical applications for precision fertilization

By systematically tracking nutrient dynamics across different growth stages, critical periods of elemental demand can be precisely identified, aligning nutrient supply with the plant's physiological needs (*Ming-Pu & San-Nai, 2006*). This approach forms the foundation of precision agriculture, enabling maximized yield and quality while enhancing resource use efficiency and promoting sustainable agricultural production (*Chen et al., 2022*; *Jat et al., 2017*; *Kumar et al., 2025*). Integrating phenological data, dry matter allocation, and the dynamic requirements of nine essential nutrients, a continuous, stage-specific fertilization strategy can be developed. This strategy aims to optimize nutrient use efficiency by synchronizing nutrient availability with the plant's real-time physiological demands,

**Table 1 Stage-specific fertilization strategy for improving the yield and quality of turmeric rhizomes and tubers.**

| Growth stage (DAP) | Phenological phase | Primary nutrient focus | Secondary/ micro-nutrient focus | Fertilization rationale & strategy |
|---|---|---|---|---|
| 0–69 | Seedling stage | Balanced N–P–K | General supply | Promote early seedling growth. A balanced basal fertilizer is the optimal choice. |
| 69–111 | Leaf rosette and initial rhizome expansion | High N, Moderate K, P | Mg, Fe | Promote vigorous shoot development to ensure sufficient photosynthesis. Nitrogen is crucial for leaf and stem development. |
| 111–139 | Tuber expansion and secondary rhizome enlargement | High K, High P, Moderate N | High Mg, Cu, Fe | Promote initial formation and enlargement of rhizomes and tubers. Potassium and phosphorus are vital for energy transfer and starch synthesis. |
| 139–195 | Dry matter accumulation in tuber and rhizome | Very High K, Moderate P | Mg, Zn | Facilitate translocation of photosynthates to storage organs. Potassium is key during this phase. Nitrogen application should be ceased. |
| 195–209 | Maturation | Cease Fertilization | None | Nutrient uptake from soil is minimal at this stage. Further fertilization provides no benefit. |

**Table 2 Key nutrient demands for high-value tuber (Yujin) development.**

| Growth stage | Macronutrients | Secondary nutrients | Micronutrients |
|---|---|---|---|
| Tubers expansion | K, P | Mg | Cu, Fe |
| Dry matter accumulation | K | Mg | Cu |
| Maturation | K | Mg | Cu |

thereby increasing both rhizome and tuber yield and quality (*Getahun, Kefale & Gelaye, 2024*). At the same time, it minimizes economic losses and negative environmental impacts such as nutrient leaching and runoff (Table 1).

In addition, this study proposes a targeted nutrient management strategy specifically for *Curcuma longa* tuber, as detailed in Table 2.

This study provides a physiological basis for optimizing turmeric yield and quality; however, several limitations remain. Notably, we did not assess the status of all micronutrients or the final nutrient profile of the soil. These aspects are planned for investigation in future studies. In addition, further research could be conducted in the following areas: (1) validating the proposed five-stage growth model and fertilization strategy across a broader range of turmeric varieties and agroecological zones; (2) examining the effects of specific micronutrients, particularly magnesium, zinc, and copper, on the accumulation of key pharmacologically active compounds such as curcumin; and (3)

performing drought and waterlogging stress experiments targeting the tubers to test the hypothesis regarding their impact on tuber development.

## CONCLUSIONS

This study systematically analyzed the dry matter accumulation and nutrient dynamics in various organs of *Curcuma longa*, identifying a nutrient demand hierarchy of potassium > nitrogen > phosphorus and defining five distinct growth stages. The results demonstrate that yield optimization requires stage-specific fertilization strategies: early-stage nitrogen application promotes the growth of aerial parts, while later stages require a focus on potassium and phosphorus to enhance the development of underground organs. This research provides a physiological foundation for precision fertilization aimed at increasing rhizome and tuber yields, improving nutrient use efficiency, and promoting a more sustainable and economically viable cultivation system.

## ACKNOWLEDGEMENTS

Special thanks to Meilin Chen, Ying Liu and Yanglin Zhan from Sichuan Zhijiacheng Biotechnology Co., Ltd. for their contributions to the field management of turmeric for this project.

### Funding

The authors received no funding for this work.

### Competing Interests

The authors declare there are no competing interests.

### Author Contributions

- Wenxin Liao conceived and designed the experiments, performed the experiments, analyzed the data, authored or reviewed drafts of the article, and approved the final draft.
- Haohan Wang conceived and designed the experiments, performed the experiments, analyzed the data, prepared figures and/or tables, authored or reviewed drafts of the article, and approved the final draft.
- Heling Fan conceived and designed the experiments, performed the experiments, authored or reviewed drafts of the article, and approved the final draft.
- Jie Chen analyzed the data, prepared figures and/or tables, and approved the final draft.
- Lili Yin performed the experiments, analyzed the data, authored or reviewed drafts of the article, and approved the final draft.
- Xiaoyang Cai analyzed the data, authored or reviewed drafts of the article, and approved the final draft.
- Min Li analyzed the data, authored or reviewed drafts of the article, and approved the final draft.

## Data Availability

   Raw data is available in the Supplemental Files.

## Supplemental Information

Supplemental information for this article can be found online at http://dx.doi.org/10.7717/peerj.19933#supplemental-information.

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
