# Peer review of "Nutrient and biomass dynamics for dual-organ yield in turmeric (Curcuma longa L.)"

_PeerJ, doi:10.7717/peerj.19933_

## Round 0.1 · original submission · Major Revisions

· Academic Editor

Major Revisions

Dear Dr. Fan,

The reviewers have highlighted several shortcomings in your manuscript. Therefore, you are advised to carefully go through their observations and comments, and modify it accordingly. You need to place more emphasis on queries relating to experimental design and procedures while revising your manuscript. Equally important are comments relating to other sections of the study. It is pertinent to mention that your revised submission will undergo additional peer review to ensure that you have revised your manuscript according to reviewers' suggestions.

Hope to receive the revised draft in due course.

**Language Note:** The review process has identified that the English language must be improved. PeerJ can provide language editing services - please contact us at [email protected] for pricing (be sure to provide your manuscript number and title). Alternatively, you should make your own arrangements to improve the language quality and provide details in your response letter. – PeerJ Staff

·

Basic reporting

The studied theme is mainly governed by the species and variety. There is a need to mention the name of a variety of turmeric taken for study.

Experimental design

There is a need to mention which basic statistical design was used for the study. There is monotony in the results, which needs to be improved with a specific common interval for the observations to be taken.

Validity of the findings

The findings are clear, but all the micronutrients need to be taken into consideration in the study.

Additional comments

The article needs improvement. The fertilizers applied should be mentioned in the materials and methods. The initial and final nutritional status of the studied area needs to be mentioned.

Reviewer 2 ·

Basic reporting

Dear Author,
I have carefully reviewed your manuscript titled “Dynamic accumulation of dry matter and nutrient demand in economic organs (rhizome and tuber) of turmeric (Curcuma longa L.) across growth stages.” The research addresses a relevant topic with practical implications; however, several key aspects need to be improved for the manuscript to meet publication standards, therefore manuscript cannot be proceed for further publication. Please consider the following suggestions for revision:
1. The current title is informative but can be refined for clarity and impact. A more concise and focused title would better reflect the core objectives and contributions of the study.
2. Clearly mention the specific growth stages evaluated in the study.
3. Briefly describe the key methodological approach used.
4. Emphasize the major outcomes and their relevance in the context of turmeric cultivation.
5. Nomenclature Consistency (Line 44): Please clarify why Curcumae radix is written in uppercase letters. Scientific names should adhere to standard conventions—typically italicized and in lowercase (except for genus).
6. The statement regarding medicinal uses—“such as promoting blood circulation, alleviating pain, and used for combat obesity, regulate inflammatory reactions, prevention of metabolic syndrome, cardiovascular problems, and diabetes”—requires more appropriate and up-to-date references to substantiate the claims.
7. The introduction is currently too brief and does not sufficiently elaborate on the background or rationale of the study.
8. Clearly articulate the objectives and highlight the novelty of the research.
9. Include a brief comparison with previous studies to demonstrate how your work addresses a specific gap.
10. Explicitly define the research gap your study aims to fill. Discuss why this research is needed and what new knowledge it contributes to the field.

Experimental design

1 Provide a citation or source for the experimental design employed, especially if it is adapted from a standard methodology.
2 Methodology Clarification: Include appropriate references for the procedures used to determine nitrogen (N), molybdenum (Mo), and potassium (K) contents. This will enhance the credibility and reproducibility of your methods.

Validity of the findings

A brief section discussing the future prospects or potential applications of the findings would add value and guide further research in this area.
Overall, the study presents interesting findings; however, it would benefit significantly from structural improvements and additional contextual depth. I encourage the authors to address the above points carefully to enhance the manuscript's scientific rigor and presentation.

·

Basic reporting

- In general, the reporting is very nice as it considers the very important agenda nutrient allocation, distribution, etc. and it should be discussed from the quality/colour of turmeric point of view.
Thare are different comments and given in general comment.

Experimental design

- Design, treatments and combination of treatments not well explained (see general comment).
- Need to be well written (clearly)

Validity of the findings

The findings are very nice, valuable, it is necessary findings/results and dicussion well syncronized

Additional comments

General Comment
• It is advisable you use, days… days after planting (DAP) in all the document body.
• Whenever working on Dynamic Accumulation of Dry Matter and Nutrient Demand in Economic Organs (Rhizome and Tuber) of Turmeric (Curcuma longa L.) Across Growth Stages, Quality of turmeric; the pigment accumulation and quantity, etc. would be very manadatory, try to include the data
• The rihzome and abouveground parts proportion, etc. need be discussed
• Your discussion part is very short, not covering all the results you presented from the findings. This need a serious attention to correct it
• On the other hand, the results are very long, advised to cluster, make a bit shorter, not very long one
Captions
• Titles of various figures; check and give appropriate titles of the figures
• Give appropriate caption of figures from bottom
• Check all figures are discussed in the text (body of the text)

Experimental Design
• Treatments; from the very beginning, all the treatments that are to be considered in the experiment need to be clearly listed.
• Design: RCBD or any other suitable designs used in this treatment need to be clearly set, with replication.

Conclusion and recommendation
• Give enough attention and discussion that satisfy the results.
• Include quality if you can and conclude and recommend exhaustively.
• Check all the references cited in texts are put in References and vice-versa

For detail of the review, please check the Ms-word main file edited with track.

---

## Round 0.2 · accepted · Accept

· Academic Editor

Accept

Dear Dr. Fan,
Thank you for your submission to PeerJ.

I am writing to inform you that your manuscript - Nutrient and biomass dynamics for dual-organ yield in turmeric ( Curcuma longa L.) - has been Accepted for publication.

Congratulations!

·

Basic reporting

I checked all the comments given in the past version of the review progress were included, justfied as commneted.

Experimental design

No comments, well justified!

Validity of the findings

No comment.

Additional comments

No additional comment.